# How a PCR Sequencing Strategy Can Bring New Data to Improve the Diagnosis of Ethionamide Resistance

**DOI:** 10.3390/microorganisms10071436

**Published:** 2022-07-15

**Authors:** Thomas Maitre, Florence Morel, Florence Brossier, Wladimir Sougakoff, Jéremy Jaffre, Sokleaph Cheng, Nicolas Veziris, Alexandra Aubry

**Affiliations:** 1Centre National de Référence des Mycobactéries et de la Résistance des Mycobactéries aux Antituberculeux, Hôpital Pitié-Salpêtrière, AP-HP (Assistance Publique Hôpitaux de Paris), Sorbonne-Université, F-75013 Paris, France; thomas.maitre@aphp.fr (T.M.); florence.morel2@aphp.fr (F.M.); florence.brossier@aphp.fr (F.B.); wladimir.sougakoff@aphp.fr (W.S.); jeremy.jaffre@aphp.fr (J.J.); sokleaph.cheng@aphp.fr (S.C.); nicolas.veziris@sorbonne-universite.fr (N.V.); 2Centre d’Immunologie et des Maladies Infectieuses, Sorbonne Université, INSERM, U1135, Cimi-Paris, F-75013 Paris, France; 3Department of Pneumology and Thoracic Oncology, Reference Centre for Rare Lung Diseases, Tenon Hospital, AP-HP, F-75020 Paris, France; 4Département de Bactériologie, AP-HP, Hôpital Saint-Antoine, Sorbonne-Université, F-75012 Paris, France

**Keywords:** ethionamide, tuberculosis, resistance, molecular diagnosis

## Abstract

Ethionamide (ETH) is a second-line antituberculosis drug. ETH resistance (ETH-R) is mainly related to the mutations of the monooxygenase-activating ETH (EthA), the ETH target (InhA), and the *inhA* promoter. Nonetheless, diagnosing ETH-R is still challenging. We assessed the strategy used for detecting ETH-R at the French National Reference Center for Mycobacteria in 497 MDR-TB isolates received from 2008 to 2016. The genotypic ETH’s resistance detection was performed by sequencing *ethA*, *ethR*, the *ethA-ethR* intergenic region, and the *inhA* promoter in the 497 multidrug-resistant isolates, whereas the phenotypic ETH susceptibility testing (PST) was performed using the reference proportion method. Mutations were found in up to 76% of the 387 resistant isolates and in up to 28% of the 110 susceptible isolates. Our results do not support the role of *ethR* mutations in ETH resistance. Altogether, the positive predictive value of our genotypic strategy to diagnose ETH-R was improved when only considering the variants included in the WHO catalogue and in other databases, such as TB-Profiler. Therefore, our work will help to update the list of mutations that could be graded as being associated with resistance to improve ETH-R diagnosis.

## 1. Introduction

The emergence of multidrug-resistant tuberculosis (MDR-TB), i.e., resistant to at least rifampin and isoniazid (INH), further threatens TB control worldwide [1]. MDR-TB remains challenging to treat, requiring second-line anti-TB drugs such as ethionamide (ETH).

Over the past several years, molecular techniques have been developed for the rapid detection of resistance to antituberculous agents since the quick detection of drug resistance is crucial for designing appropriate antituberculosis drug regimens, preventing treatment failure and/or relapse, and reducing the spread of drug-resistant isolates. Molecular assays for the detection of mutations related to resistance have been increasingly used and have led to shortening the time to detection of resistance to one working day [2]. 

Ethionamide (ETH) is a derivative of isonicotinic acid that is structurally similar to INH. ETH and INH are pro-drugs requiring activation by different pathways: the KatG catalase-peroxidase for INH and the EthA monooxygenase (negatively regulated by EthR) for ETH [3]. ETH and INH have a common target: the enoyl-ACP reductase InhA. EthA is a NADPH-specific flavin adenine dinucleotide-containing monooxygenase and a Baeyer–Villiger monooxygenase and is involved in cell wall biosynthesis [4,5]. Previous studies have reported that mutations of *ethA* and *inhA* are the main mutations reported in ETH-R strains: *ethA* mutations caused 37% to 100% of ETH-R in *M. tuberculosis*, and mutations in *inhA* caused 25% to 100% of ETH-R in *M. tuberculosis*, whereas mutations in EthR and the *inhA* promoter were less frequent [6,7].

The increasing use of genotypic diagnosis of resistance in tuberculosis management requires the extensive study and classification of mutations identified in genes involved in drug resistance, i.e., genotypic/phenotypic correlation. For ETH, only 24 mutations were confidently graded by the WHO as being associated with resistance in *ethA*, *inhA,* and the promoter region of *inhA*, which stresses the need to add new data enabling an update to that list (without taking into account the mutations graded as “uncertain significance” that are listed in the extended catalogue [8]). 

Therefore, the present study aimed to investigate the prevalence of mutations in *ethA*, *ethR*, the *inhA* gene, and the promoter region of *inhA* associated with independent resistance to ethionamide (ETH-R) in *M. tuberculosis* isolates based on a prospective genotyping strategy used at the French National Reference Center (NRC) for Mycobacteria to diagnose ETH-R.

## 2. Materials and Methods

### 2.1. Mycobacterium tuberculosis Complex Clinical Isolates 

A total of 497 MDR *M. tuberculosis* complex isolates from TB cases diagnosed in France and received at the French National Reference Center for Mycobacteria were collected over a 9-year (2008–2016) period. On the basis of the results of DST, the 497 MDR isolates were classified as ETH-R (*n* = 387) or as susceptible to ETH (ETH-S) (*n* = 110) (Table 1).

### 2.2. Phenotypic Drug Susceptibility Testing and Quality Controls

DST was performed on Löwenstein–Jensen medium using the reference standard proportion method [9] and concentrations of 40 mg/L for ETH [10,11]. Resistance to ETH was defined as a proportion of resistant mutants ≥ 1% at 40 mg/L [10,11]. Quality controls were performed for each new batch of LJ medium containing ETH using the reference strain *M. tuberculosis* H37Rv. The laboratory underwent external quality validation through the External Quality Assurance (EQA) systems, ensuring the accurate diagnosis of TB and drug-resistant TB through the European TB reference laboratory network (ERLTB-Net) organized by INSTAND.

### 2.3. DNA Sequencing of Genes Associated with Ethionamide Resistance 

For all of the MDR-TB isolates received at the French NRC, the entire *ethA* and *ethR* genes, and the *ethA-ethR* intergenic region were prospectively sequenced, and mutations in the *inhA* promoter region were determined by using the Genotype MTBDRplus test (Hain Lifescience GmbH, Nehren, Germany) according to the manufacturer’s instructions. For the isolates in which wild-type *ethA* and *inhA* promoter were observed, the entire *inhA* gene and its promoter were retrospectively sequenced in ETH-R isolates. *EthA*, *ethR*, the *ethA/ethR* intergenic region, and *inhA* and its promoter were amplified and sequenced using the primers previously described [6]. Genomic DNA was isolated from bacteria grown on Lowenstein–Jensen medium. A loop of culture was resuspended in water (500 µL) and inactivated by heating at 95 °C for 15 min. DNA (5 µL) obtained by heat shock extraction (1 min at 95 °C and 1 min in ice, repeated five times) was used for PCR amplification with the following steps: denaturation of 5 min at 95 °C followed by 35 cycles of 1 min at 95 °C, 1 min at the primer-dependent annealing temperature (*Ta*), 1 min at 72 °C, and a final extension step of 7 min at 72 °C. The PCR products were purified by filtration with Microcon 100 microconcentrators (Amicon Inc., Beverly, MA, USA), and the PCR for Sanger sequencing was conducted using a BigDye Terminator cycle sequencing ready kit (Applied Biosystems, Courtaboeuf, France).

### 2.4. Databases

The impact of the mutations on ETH-R was evaluated by looking for those graded as being associated with resistance in the WHO catalogue. For those not graded as being associated with ETH-R in the catalogue, we checked if they were listed in other published databases, such as TB-Profiler [12,13], PhyResSE [14], and the one published by Manson et al. [15].

### 2.5. Statistical Analysis

The proportion of the ETH-R and ETH-S isolates with the different mutations were compared using Fisher’s exact test. The *p* values were two-tailed, and *p* values of ≤0.05 were considered significant. Sensitivity, specificity, positive predictive value, and negative predictive value of the prospective genotypic strategy used at the French NRC were determined by considering the phenotypic DST as the reference standard. Since the entire *inhA* gene was only sequenced retrospectively in ETH-R isolates displaying wild-type *ethA* and the *inhA* promoter, it is not considered as part of the prospective genotypic strategy. 

## 3. Results

Among the 497 MDR-TB isolates, 78% (*n* = 387) were classified as ETH-R and 22% (*n* = 110) as ETH-S by phenotypic DST (Table 1 and Table 2). A total of 123 mutations were evidenced in *ethA*, *ethR*, the *ethA/ethR* intergenic region, and *inhA* and its promoter. 

Most of the alterations observed had never been reported, no matter their susceptibility to ETH [6,16,17,18,19], and only 6 (5%) were cited as being associated with resistance in the WHO catalogue [20], and 20 additional mutations among the 123 (17%) were cited in published databases of drug resistance mutations (Table 1 and Table 2). Altogether, among the 387 ETH-R isolates, 294 (76%) had at least one mutation in *ethA*, *ethR*, the *ethA-ethR* intergenic region, and the *inhA* promoter, and 93 (24%) had no mutation, whereas among the 110 ETH-S isolates, 79 (72%) had no mutations in *ethA*, *ethR*, the *ethA-ethR* intergenic region, and the *inhA* promoter, and 31 (28%) had at least one mutation (*p* < 10^−5^) (Table 1). 

Among the 123 mutations evidenced in *ethA*, *ethR*, the *ethA/ethR* intergenic region, and *inhA* and its promoter, 4 corresponded to phylogenetic SNPs: G124D, S266R, and 768_del_g in EthA and V78A in InhA (Table 3) [21], and therefore, they have no impact on ETH susceptibility. 

Among the 123 mutations, 112 (91%) were only evidenced in the ETH-R strains. Among these 112 mutations, after the exclusion of known polymorphisms, 61 (54%) were the only mutations evidenced, and no other mutation was found in the candidate genes. Among the 112 different mutations that were only observed in ETH-R strains, almost half (*n* = 57.5%) are reported as being associated with ETH-R in the WHO catalogue and/or other databases (Table 1). 

Interestingly, the mutations suspected to have an important impact on protein function (i.e., the introduction of a stop codon and large deletions) were not evidenced at all in the ETH-S strains (Table 2).

By comparing the proportions of the mutated strains in the ETH-R and ETH-S strains per gene, the difference was statistically significant for *ethA* (197/387 (51%) versus 28/110 (25%), *p* < 10^−5^), the *inhA* promoter (139/387 (36%) versus 2/110 (2%), *p* < 10^−5^), and the *ethA-ethR* intergenic region (15/387 (4%) versus 0/110 (0%), *p* = 0.035), but not for *ethR* (13/387 (3%) versus 3/110 (3%), *p* = 0.74) (Table 1 and Table 2).

By using the phenotypic data as the reference, the genotyping strategy used in the French National Reference Center (NRC) for Mycobacteria had 78.8% sensitivity to diagnose ETH-R, 76.4% specificity, 92.0% positive predictive value (PPV), and 49.4% negative predictive value (NPV) (with the exclusion of known phylogenetic SNPs). Modifying this strategy by deleting ethR mutations whose role in ETH-R is unclear [6,22,23] and did not modify its performance (Table 3), whereas only considering the mutations mentioned in the WHO catalogue dramatically increased the PPV performance [20]. Interestingly, when combining the interpretation of the impact of the mutations and the published databases to the WHO catalogue, the sensitivity increased dramatically (36.7% vs. 52.9%), but the PPV of the genotyping strategy used to predict ETH-R only slightly increased at the expense of the NPV value (Table 3).

## 4. Discussion

The old antibiotic ETH, which was used for a long time to treat drug-resistant TB [1], is benefiting from a renewal since the development of new compounds boosting its activity in vivo [24,25] are currently being assessed in a Phase 1 trial (ClinicalTrials.gov n° NCT04654143). 

The increase in the proportion of ETH-R isolates among the MDR-TB isolates received at the French NRC from 2008–2009 (44%) to 2010–2016 (77%) represents a challenge to the existing health care facilities for the management of MDR-TB and XDR-TB who follow programmatic regimens, and this underscores the need for reliable methods for ETH DST [26].

The previously published studies aiming at deciphering the molecular bases of resistance to ETH [6,27,28,29,30,31,32] had some limitations: Firstly, most of the studies were performed on retrospectively chosen isolates, either ETH-R isolates exclusively or in a high prevalence of ETH-R isolates. Secondly, no study included the sequencing of all of the main genes described as being implicated in ETH-R (i.e., *ethA, ethR*, the *ethA*-*ethR* intergenic region, and *inhA* and its promoter). Therefore, our study prospectively evaluated the performance of a genotypic strategy based on the sequencing of the main genes known to be involved in ETH-R to diagnose ETH-R, which brings useful light into the field, especially as whole genome sequencing becomes widely used, making it even more complex to diagnose the resistance of a drug such as ETH. 

Overall, the performance of our genotypic strategy to diagnose ETH-R had suboptimal performance (Table 3). Interestingly, the performance of our genotypic strategy was enhanced by combining the interpretation of the mutations based on the data from the WHO catalogue and other databases (Table 3), as illustrated by the improvement in the specificity (98.2% vs. 76.4%) and the PPV (99.0% vs. 92.0%). Since the role of *ethR* mutations in ETH-R is doubtful, we propose to avoid taking these mutations into account to diagnose ETH-R [6,22,23]. Our genotypic strategy, when either taking the *ethR* mutations into account or not, did not modify the diagnostic performance, confirming the probable lack of involvement in the resistance of *ethR* mutations (Table 3).

In light of these results, carefully analyzing the strains without genotypic/phenotypic correlation is of great interest. The 26 ETH-S isolates with at least one alteration, phylogenetic SNPs excluded (Table 2), may be explained by the fact that (i) the genetic alteration observed is not implicated in ETH-R, (ii) the strains have a low level of ETH-R which is not detected by the phenotypic DST [33,34,35,36], or (iii) compensatory mutations restore ETH susceptibility. Other flavin monooxygenases such as EthA2 or MymA are involved in ETH activation [24,37]. These alternative activation pathways may compensate for the impact of mutations in EthA that are suspected to provide a loss of protein function (i.e., deletion or insertion) by restoring susceptibility to ETH despite the presence of an EthA mutation (Table 2). It is worth noticing that no mutations that are suspected to provide an important loss of protein function (stop codon and large deletions) were observed in the ETH-S strains (Table 2). The 96 ETH-R isolates without any mutations (phylogenetic SNPs excluded) may be explained by the fact that (i) mutations are present in genes other than those studied (*mshA*, *Rv3083*, *ndh*, *Rv0565c* [38,39]), and (ii) the strains were wrongly classified as resistant by the phenotypic DST (see below). 

Overall, both false results questioned the ability of genotyping, but also of phenotypic DST, to properly classify strains as susceptible or resistant. The challenges associated with *M. tuberculosis* DST are well known, especially for ETH. First, the drug is thermolabile, which makes DST difficult [33]. Second, discriminating ETH-resistant and -susceptible strains can be a challenge since the distribution of their Minimum Inhibitory Concentrations (MICs) partially overlap [34]. This is similar to what has been described for ethambutol, supporting the idea that the reporting of strains with MICs close to the ECOFF could be affected by reproducibility issues, as the classification into susceptible or resistant highly depends on methodological variation [34]. Third, ETH DST has been shown to have poor concordance and reproducibility compared to other drugs [35,36]. As previously suggested, creating an intermediate susceptibility classification for ETH-S strains with gene alteration, supported by the unfavorable pharmacodynamic indices, could be warranted to increase reproducibility and to account for methodological variation [34].

Most of the alterations described in our study (about half of the mutations exclusively found in ETH-R strains) had never been described before, and little is currently known about the effects of the different mutations found in *ethA*, *ethR,* and the *ethA-ethR* intergenic region. Therefore, we have provided new data that can contribute to enriching the listed mutations that are associated with ETH-R in the WHO catalogue and other databases. 

## Figures and Tables

**Table 1 microorganisms-10-01436-t001:** Mutations in *ethA*, *inhA* and its promoter, *ethR*, and *ethA-ethR* intergenic region for the 387 ETH-R MDR-TB isolates.

N° of Isolates	Sequencing Results ^a^				
	*ethA*	*inhA* Promoter	*inhA*	*ethR*	*ethA-ethR* Intergenic Region
1	** M1R **	wt	np	wt	wt
2	G11S	wt	np	wt	wt
2	S15P	** c-15t **	np	wt	wt
1	A19V	**g-17t**	np	wt	wt
1	**H22P**	wt	np	wt	wt
1	**H22P**	** c-15t **	np	wt	wt
3	**H22P**	** c-15t **	np	**F110L**	wt
1	C27W	wt	np	wt	wt
1	C27W	**g-17t**	np	wt	wt
1	G36D	wt	np	wt	wt
1	G42V, P334A	wt	np	wt	wt
1	F48S	wt	np	wt	wt
1	Y50C	wt	np	wt	wt
1	S55C	wt	np	wt	wt
1	F66L, G299D	wt	np	wt	wt
1	G78D	** c-15t **	np	wt	wt
1	A89E, *S266R*	wt	np	wt	wt
1	A89E, R99Q, *S266R*	wt	np	wt	a-9g
1	D95N	wt	np	wt	wt
1	D95N	** c-15t **	** S94A **	wt	wt
1	D95N	** c-15t **	np	wt	wt
1	W109 !	wt	np	wt	wt
1	G124D	wt	np	wt	wt
2	L136R	** t-8a **	np	wt	wt
2	C137R	wt	np	wt	wt
1	G139D	wt	np	wt	wt
2	Y140 !	wt	np	wt	wt
1	Y141C, 1367_ins_7nt	wt	np	M142I, Q143K	wt
1	** Y147 ! **	wt	np	wt	wt
6	**Q165P**	wt	np	wt	wt
2	W167G	wt	np	wt	wt
1	S183R	** c-15t **	np	wt	wt
1	P192S	** c-15t **	np	wt	wt
1	P192T	wt	np	wt	wt
1	V202G	wt	np	wt	wt
1	**Q206 !**	wt	np	wt	wt
2	S208 !	wt	np	wt	wt
1	Y211S	** c-15t **	np	wt	wt
1	**E223K**	** c-15t **	np	wt	wt
1	N226D	wt	np	wt	wt
3	V238G	wt	np	wt	wt
1	R239L	wt	np	wt	wt
1	Q254P	** c-15t **	np	wt	wt
1	W256 !	wt	np	wt	wt
4	P257S	** c-15t **	np	wt	wt
3	*S266R*	wt	np	wt	wt
1	*S266R*	wt	np	D23G	wt
9	**Q269 !**	** c-15t **	np	wt	wt
1	**Q269 !**	** c-15t **	np	wt	a-9g
1	**L272P**	wt	np	wt	wt
4	H281P	** c-15t **	np	wt	wt
1	C294Y	** c-15t **	np	wt	wt
1	I305N	wt	np	wt	wt
2	T314I	wt	np	wt	wt
2	T314I	** c-15t **	np	wt	wt
1	T314I	** t-8c **	np	wt	wt
1	I337V	WT	*V78A*	wt	wt
3	**I338S**	** c-15t **	np	wt	wt
1	**T342K**	wt	np	wt	wt
1	M372R	wt	np	wt	wt
2	N379D	wt	np	wt	wt
1	**G385D**	wt	np	wt	wt
2	C403R	wt	np	wt	wt
2	P422L	wt	np	wt	wt
2	L440P	wt	np	wt	wt
1	Q449R	wt	np	wt	wt
1	D464G	wt	np	wt	wt
1	R471P	** c-15t **	np	wt	wt
1	R483T	wt	np	wt	wt
1	32_del_g	wt	np	wt	wt
1	57_ins_4nt	** t-8c **	np	wt	wt
1	109_del_a	** t-8c **	np	wt	wt
19	**110_del_a**	wt	np	wt	wt
1	**110_del_a**	** t-8c **	np	wt	wt
1	137_del_a	** t-8c **	np	wt	wt
1	328_ins_t	wt	np	wt	wt
1	373_ins_a	** c-15t **	np	wt	wt
1	390_del_c	wt	np	wt	wt
1	437_ins_g	wt	np	wt	wt
1	477_del_g	wt	np	wt	wt
1	509_del_a	wt	np	M102T	wt
1	522_del_c	wt	np	wt	wt
1	537–790_del	wt	np	wt	wt
1	626_del_cc	wt	np	wt	wt
3	**639_del_gt**	wt	np	wt	wt
5	**703_del_t**	wt	np	wt	wt
1	751_del_a	wt	np	wt	wt
1	752_ins_g	wt	np	wt	wt
9	**768_del_g**	wt	np	wt	wt
1	778_del_a	** c-15t **	np	wt	wt
2	831–837_del	** c-15t **	np	wt	wt
4	**884_del_t**	wt	np	wt	wt
1	935_ins_t	wt	np	wt	wt
1	1010_del_t	wt	np	wt	wt
1	1034_del_a	wt	np	wt	wt
1	**1054_del_g**	** t-8c **	np	wt	wt
1	1061_ins_c	** c-15t **	np	wt	wt
1	1222_del_t	wt	np	wt	wt
6	**1242_del_t**	wt	np	wt	wt
1	1281_ins_a	wt	np	wt	wt
1	1292_del_t	wt	np	wt	wt
1	1292_del_t	wt	np	wt	a-9g
1	1343_del_a	** c-15t **	np	wt	wt
1	1391_ins_a	** c-15t **	np	wt	wt
1	1431_ins_t	WT	np	wt	wt
1	1466_del_tt	** c-15t **	np	wt	wt
5	1470_del_g	wt	np	wt	wt
1	**933–1737_del**	wt	np	wt	a-40g
3	**large deletion**	wt	np	**large deletion**	**large deletion**
1	**large deletion**	wt	np	wt	wt
1	wt	wt	** S94A **	wt	wt
1	wt	wt	** S94A **	wt	wt
1	wt	wt	** S94A **	wt	wt
8	wt	wt	wt	wt	a-68g
1	wt	wt	wt	T149A	wt
1	wt	wt	wt	S131R	wt
1	wt	wt	wt	M142I, Q143K	wt
1	wt	wt	wt	P195L	wt
82	wt	** c-15t **	np	wt	wt
93	wt	wt	WT	wt	wt

Mutations are indicated as amino acids for all proteins encoded by the corresponding genes except for the inhA promoter and the ethA-ethR intergenic region, for which mutations are indicated in nucleotide. Mutations reported in the WHO catalogue as being associated with ETH-R are bolded and underlined. Mutations not listed in the WHO catalogue but mentioned in other published databases are bolded. Phylogenetic SNPs are indicated in italics. Stop codon is represented with “!”. ^a^ wt: wild-type; mut: mutated, np: not performed.

**Table 2 microorganisms-10-01436-t002:** Mutations in *ethA*, *inhA* and its promoter, *ethR*, and *ethA-ethR* intergenic region for the 110 ETH-S MDR-TB isolates.

N° of Isolates	Sequencing Results ^a^				
	*ethA*	*inhA* Promoter	*inhA*	*ethR*	*ethA-ethR* Intergenic Region
2	I9T	wt	np	wt	wt
1	G11D	wt	np	wt	wt
2	D95N, *768_del_g*	wt	np	wt	wt
1	C131Y	wt	np	wt	wt
2	W167S, *S266R*	wt	np	S131R	wt
1	I178S	wt	np	wt	wt
3	*S266R*	wt	np	wt	wt
1	C294Y	wt	np	wt	wt
1	T314I	wt	np	wt	wt
1	P334A	wt	np	wt	wt
1	N379D	wt	np	wt	wt
1	110_del_a	wt	np	wt	wt
1	382_ins_g	wt	np	wt	wt
1	626_del_cc	wt	np	wt	wt
3	703_del_t	wt	np	wt	wt
2	*768_del_g*	wt	np	wt	wt
1	851_ins_c	wt	np	wt	wt
1	935_ins_t	wt	np	wt	wt
1	1034_del_a	wt	np	wt	wt
1	1242_del_t	wt	np	wt	wt
1	wt	wt	np	65_ins_cg	wt
2	wt	** c-15t **	np	wt	wt
79	wt	wt	np	wt	wt

Mutations are indicated as amino acids for all proteins encoded by the corresponding genes except for the inhA promoter and the ethA-ethR intergenic region for which mutations are indicated in nucleotide. Mutations reported in the WHO catalogue as being associated with ETH-R are bolded and underlined. Mutations not listed in the WHO catalogue but mentioned in other published databases are bolded. Phylogenetic SNPs are indicated in italics. ^a^ wt: wild-type; mut: mutated, np: not performed.

**Table 3 microorganisms-10-01436-t003:** Performances of the sequencing strategy used to diagnose ETH-R (i.e., PCR sequencing of *ethA*, *inhA* and its promoter, *ethR*, and *ethA-ethR* intergenic region) regarding the criteria used to interpret the results, compared to the phenotypic DST as a gold standard.

Criteria Used to Interpret the Mutations	Sensitivity	Specificity	PPV ^a^	NPV ^b^
none ^c^	78.6	71.8	90.7	48.8
none, ethR mutations excluded	74.9	72.7	90.6	45.2
none, polymorphisms excluded	77.8	76.4	92	49.4
none, polymorphisms and ethR mutations excluded	75.2	76.4	91.8	46.7
WHO catalogue only	36.7	98.2	98.6	30.6
WHO catalogue + databases	53.2	98.2	99.0	37.4
WHO catalogue + databases (ethR mutation excluded)	52.9	98.2	99.0	37.2

^a^ PPV: positive predictive value; ^b^ NPV: negative predictive value; ^c^ all mutations are taken into account.

## Data Availability

The data presented in this study are available on request from the corresponding author.

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
