# Peer review of "How a PCR Sequencing Strategy Can Bring New Data to Improve the Diagnosis of Ethionamide Resistance"

_microorganisms, 2022, doi:10.3390/microorganisms10071436_

Round 1
Reviewer 1 Report
Line 90
The isolates do not actively participate in displaying. The wild type sequences were observed.
Line 173
reported
Line 214
The word should probably be ilogical
Author Response
-Line 90, The isolates do not actively participate in displaying. The wild type sequences were observed.
Answer: the sentence has been modified (line 101 in the revised version of the manuscript).
-Line 173, reported
Answer: the word has been modified (lines 180 and 191 in the revised version of the manuscript).
-Line 214, The word should probably be illogical
Answer: the sentence has been modified (line 228 in the revised version of the manuscript).
Reviewer 2 Report
REVIEW
ABSTRACT
Lines 26-27: this expression is unclear and paragraph might be clarified, for example: “Mutations were found in up to 76% of 387 resistant isolates and in up to 28% of 110 susceptible isolates”
INTRODUCTION
Lines 49-52: this paragraph is misleading: “previous studies reported that mutations of ethA caused 37% of ETH-R in M.tuberculosis, whereas mutations in inhA caused 25% of ETH-R in M. tuberculosis”
Lines 54-56: the paragraph´s redaction is misleading. Please, clarify.
M&M
Table 1: is too long. Data analysis can improve if strains could be grouped in a different manner, for example: all strains exhibiting mutations with fenotypic resistance (ETH-R in agar proportion method) in a single table. The rest of strains (ETH-S) in a separate table showing their respective mutations.
Table 2: The real significance is unclear: the reader doesn´t understand what does it mean? Which is the clinical interest of this PCR-sequencing assay?
RESULTS
Lines 121-123: this paragraph is hard to read: “Of the 497 MDR-TB isolates, 78% (N=387) isolates were clasified as ETH-R and 22% (n=110) were clasified as ETH-S”.
In 24% of 387 ETH-R there are no mutations: Why these strains are ETH-R by proportion method? How to explain these data?
In 28% of 110 ETH-S there are one mutation: can all of these mutations develop phenotypical resistance in these isolates?
DISCUSSION
Lines 194-196: Please, realign the words to simplify this paragraph.
Lines 202-203: point (i) focuses in the same limitations of this study , and this note is not convenient.
Lines 205-209: to assess this conclusion, the authors must define previously clear results about the real usefullness of the proposed PCR assay.
Lines 214-16: If so, why to detect all of these ETH mutations? Which is the clinical usefullness? The authors must define clearly which mutations are clue to detect ETH resistance. I don´t understand the final purpose of this complex PCR assay.
Lines 219-220: These aren´t false positive, these are probably not expressing mutations. Really, is most correct to express this concern as: correlation between fenotypic assay vs genotypic assay.
Lines 223-229: Perhaps this problem is related to the long time for recovering all of isolates: different culture media, variation of ETH provider, exact method to determine ETH resistance in the Reference Laboratory…
Lines 231-241: Accurate reading of this paragraph restate all the study: if ETH-R gold standard is based on proportion method, is gathering all these methodological variation. Really, to select this as gold standard, authors must repeat ETH susceptibility testing of all of isolates under the same requirements (the same ETH, the same culture media and method…).
Author Response
ABSTRACT
-Lines 26-27: this expression is unclear and paragraph might be clarified, for example: “Mutations were found in up to 76% of 387 resistant isolates and in up to 28% of 110 susceptible isolates”
Answer: the sentence has been modified according to the reviewer suggestion (lines 27 and 28 in the revised version of the manuscript).
INTRODUCTION
-Lines 49-52: this paragraph is misleading: “previous studies reported that mutations of ethA caused 37% of ETH-R in M. tuberculosis, whereas mutations in inhA caused 25% of ETH-R in M. tuberculosis”
Answer: the sentence has been modified according to the reviewer suggestion, however the rank of reported mutations in each gene has been added to not be misleading (lines 54-57 in the revised version of the manuscript).
-Lines 54-56: the paragraph´s redaction is misleading. Please, clarify.
Answer: we are note sure of the meaning of reviewer’s remark. However, the paragraph has been modified (lines 62-64 in the revised version of the manuscript) and we hope it will be more suitable to the reviewer as it is now. The increasing use of genotypic diagnosis of resistance in tuberculosis management requires an extensive study and classification of mutations identified in genes involved in drug resistance, i.e. genotypic/phenotypic correlation. For ETH, only 24 mutations were confidently graded by WHO as associated with resistance in ethA, inhA and the promoter region of inhA, stressing out the need to add new data enabling to update that list (without taking into account the mutations graded as “uncertain significance” listed in the extended catalogue [8]).”
M&M
-Table 1: is too long. Data analysis can improve if strains could be grouped in a different manner, for example: all strains exhibiting mutations with phenotypic resistance (ETH-R in agar proportion method) in a single table. The rest of strains (ETH-S) in a separate table showing their respective mutations.
Answer: as requested table 1 has been splitted into 2 different tables, one presenting the ETH-R data, the other one, the ETH-S data.
-Table 2: The real significance is unclear: the reader doesn´t understand what does it mean? Which is the clinical interest of this PCR-sequencing assay?
Answer: to improve the significance of this table, the title has been modified as well as the labelling of the first column of the table (lines 199-203 in the revised version of the manuscript).
RESULTS
-Lines 121-123: this paragraph is hard to read: “Of the 497 MDR-TB isolates, 78% (N=387) isolates were classified as ETH-R and 22% (n=110) were classified as ETH-S”.
Answer: the sentence has been modified according to the reviewer suggestion (lines 134-135 in the revised version of the manuscript).
-In 24% of 387 ETH-R there are no mutations: Why these strains are ETH-R by proportion method? How to explain these data?
Answer: this very important point raised by the reviewer is discussed in the discussion section (lines 253-264 in the revised version of the manuscript). Overall, this questions the ability of genotyping, but also phenotypic DST, to properly classify strains as susceptible or resistant. The challenges associated to M. tuberculosis DST are well known, especially for ETH.
-In 28% of 110 ETH-S there are one mutation: can all of these mutations develop phenotypical resistance in these isolates?
Answer: this percentage illustrates the fact that all mutations occurring in genes involved in antibiotic mode of action of a drug are not necessarily involved in drug resistance. Among the different mutations evidenced in ETH-S strains, 11 were only identified in ETH-S strains strongly suggesting that they are not involved in drug resistance whereas 12/23 were identified in both ETH-S and ETH-R strains suggesting that this mutation could either be responsible for a low level of resistance or be a polymorphism. This very important point raised by the reviewer is also discussed in the discussion section (lines 236-244 in the revised version of the manuscript).
DISCUSSION
-Lines 194-196: Please, realign the words to simplify this paragraph.
Answer: the sentence has been modified (lines 208-210 in the revised version of the manuscript).
-Lines 202-203: point (i) focuses in the same limitations of this study, and this note is not convenient.
Answer: the sentence has been modified (lines 216 and 218 in the revised version of the manuscript).
-Lines 205-209: to assess this conclusion, the authors must define previously clear results about the real usefulness of the proposed PCR assay.
Answer: the usefulness of the proposed PCR strategy has been clarified, especially in the Table 3.
-Lines 214-16: If so, why to detect all of these ETH mutations? Which is the clinical usefullness? The authors must define clearly which mutations are clue to detect ETH resistance. I don´t understand the final purpose of this complex PCR assay.
Answer: one of the objectives of this study was to help to understand the implication of mutations in ETH-R. Our work confirms that mutations in ethA-ethR intergenic region are mainly involved in ETH-R whereas mutations in ethR seems equally presented in both ETH-R and ETH-S strains. However, the scope of our work was not to demonstrate the impact of these mutations in susceptibility or resistance to ETH; therefore, further studies are needed to decipher the role of each mutation in ETH-R.
-Lines 219-220: These aren´t false positive, these are probably not expressing mutations. Really, is most correct to express this concern as: correlation between fenotypic assay vs genotypic assay.
Answer: the sentence has been modified as suggested (lines 236 and 245 in the revised version of the manuscript).
-Lines 223-229: Perhaps this problem is related to the long time for recovering all of isolates: different culture media, variation of ETH provider, exact method to determine ETH resistance in the Reference Laboratory…
Answer: over the years there were no changes in the reference of culture media and ETH provider; also the same method was used together with internal quality control and participation to EEQ where good results were obtained. Therefore, this may be an explanation, but it seems less likely than the reasons discussed in the discussion section (the drug is thermolabile, which makes DST difficult; discriminating ETH-resistant and -susceptible strains can be a challenge since the distribution of their Minimum Inhibitory Concentrations (MICs) overlap partially).
-Lines 231-241: Accurate reading of this paragraph restate all the study: if ETH-R gold standard is based on proportion method, is gathering all these methodological variation. Really, to select this as gold standard, authors must repeat ETH susceptibility testing of all of isolates under the same requirements (the same ETH, the same culture media and method…).
Answer: drug susceptibility testing (DST) of M. tuberculosis strains is historically based on the proportion method using Löwenstein-Jensen medium and concentrations of 40 mg/liter for ETH [10,11]; resistance to ETH being defined as a proportion of resistant mutants ≥1% [10,11]. To avoid variations that could be due to factors such those mentioned by the reviewer, internal quality controls are performed for each new batch of LJ containing ETH by using the reference strain M. tuberculosis H37Rv. As said above, the laboratory also successfully participates to the External Quality Assurance (EQA) systems ensuring accurate diagnosis of TB and drug-resistant TB organized by INSTAND. Therefore, we are confident in our results.